# Scalable interpolation of satellite altimetry data with probabilistic machine learning

William Gregory [1] ✉, Ronald MacEachern[2,3], So Takao[2], Isobel R. Lawrence[4], Carmen Nab [3,5], Marc Peter Deisenroth[2,6] & Michel Tsamados [3]

We present GPSat; an open-source Python programming library for performing efficient interpolation of non-stationary satellite altimetry data, using scalable Gaussian process techniques. We use GPSat to generate complete maps of daily 50 km-gridded Arctic sea ice radar freeboard, and find that, relative to a previous interpolation scheme, GPSat offers a 504 × computational speedup, with less than 4 mm difference on the derived freeboards on average. We then demonstrate the scalability of GPSat through freeboard interpolation at 5 km resolution, and Sea-Level Anomalies (SLA) at the resolution of the altimeter footprint. Interpolated 5 km radar freeboards show strong agreement with airborne data (linear correlation of 0.66). Footprint-level SLA interpolation also shows improvements in predictive skill over linear regression. In this work, we suggest that GPSat could overcome the computational bottlenecks faced in many altimetry-based interpolation routines, and hence advance critical understanding of ocean and sea ice variability over short spatio-temporal scales.

Earth observation satellites have made it possible to monitor virtually the entirety of the Earth's surface in relatively short periods of time. This has accelerated our understanding of weather and climate processes, and the rate and scale to which these processes are impacted by anthropogenic climate change. Satellite-mounted altimeters, in particular, play a crucial role in this effort by recording changes in the elevation of both ocean and sea ice surfaces. For example, the TOPEX/Poseidon altimeter (Topography Experiment - Positioning, Ocean, Solid Earth, Ice Dynamics, Orbital Navigator) allowed major breakthroughs in tracking global sea-level rise[1,2], through higher precision sea-surface height estimates[3]. Furthermore, polar-monitoring altimeters, such as CryoSat-2, have provided a consistent record of sea ice thickness changes over the past decade[4–7]. Sea ice thickness plays a crucial role in the climate system; as a major control on atmosphere-ocean heat exchanges[8], sea ice teleconnections[9,10], the timing of ice-algal blooms[11], and the overall response of sea ice to global warming[12]. Local sea ice thickness variations are governed by a myriad of dynamic and thermodynamic

factors, but typical values range from a few centimetres to a few metres.

Altimeters record surface elevation by the return time of an emitted microwave pulse (radar altimeter) or laser beam (laser altimeter), collecting data along narrow tracks as they orbit the Earth. The horizontal resolution, and hence the ability to resolve small-scale features of the sea ice and ocean, is controlled by the altimeter footprint; the surface area covered by a single microwave pulse or laser beam. A larger footprint means that the altimeter can survey the Earth's surface more quickly, although at the cost of horizontal resolution. Therefore, depending on the specific altimeter and zone of interest, observations can be inherently sparse on timescales ranging from days to weeks. For example, ICESat-2 (Ice, Cloud, and Land Elevation Satellite) is a laser altimeter carrying three pairs of beams, each with an along-track footprint size of ~17 m[13]. Meanwhile, CryoSat-2 is a radar altimeter with an along-track footprint size of ~300 m[14]. Importantly, both altimeters require approximately 30 days to uniformly sample the sea ice cover at both poles (at the typical grid resolution used in polar altimetry[15]). This

[1]Atmospheric and Oceanic Sciences Program, Princeton University, Princeton, NJ, USA. [2]UCL Centre for Artificial Intelligence, University College London, London, UK. [3]Centre for Polar Observation and Modelling, University College London, London, UK. [4]ESRIN, European Space Agency, Frascati, Italy. [5]Ocean Forecasting Research & Development, Met Office, Exeter, UK. [6]The Alan Turing Institute, London, UK. ✉e-mail: wg4031@princeton.edu

poses limitations in our ability to understand the processes that drive ocean and sea ice variability on timescales ranging from days to weeks, as well as hindering applications which would otherwise benefit from high spatio-temporal resolution observations. Such applications include initialising numerical weather prediction and/or climate models[16], and Arctic maritime navigation[17]. In these cases, high spatio-temporal sea ice thickness observations could provide more utility than traditional area-based quantities. For example, sea ice thickness strongly controls summer sea ice melt rates, therefore accurate thickness initialisation will allow more faithful representation of sea ice evolution in summer than by just initialising the ice extent. Furthermore, accurate sea ice thickness conditions for shipping forecasts can provide safe passage for icebreakers traversing through the ice—information which cannot be inferred from sea ice concentration.

To overcome the inherent sparsity in satellite altimetry data, many studies have leveraged statistical interpolation schemes to effectively fill the gaps at unobserved locations. Such schemes have been deployed under a wide variety of names in the Earth observation literature, including optimal interpolation[18–23], objective analysis[24–26], kriging[27,28], and Gaussian process (GP) regression[29]. Each of these approaches can be considered variations of linear smoother models, which are a broad class of Bayesian inverse methods that rely on covariance-based weighting to make predictions. Common to all of these applications is the poor scaling of the methodology to large data sets (typically over a few thousand data points), owing to the computation of matrix inverses, which scales cubically in the size of the data set. This often means that studies must rely on parallelised high-performance computer (HPC) environments or data sub-sampling to generate predictions in a reasonable time. In some cases, such computational restrictions prevent potentially novel data products from becoming fully operational and open-source. Landy et al.[26], for example, explored the spatial interpolation of along-track CryoSat-2 sea-surface height profiles in the Arctic. In their case, even with significant sub-sampling of the training data, 96 hours were still required to process a single month of data. Similarly, Gregory et al.[29], hereafter G21, investigated the joint spatio-temporal interpolation of CryoSat-2, Sentinel-3A and Sentinel-3B gridded sea ice freeboard observations (that is, the height of a sea ice floe relative to the underlying ocean surface). With the use of a parallelised HPC environment (25 CPU cores), 36 h were required to process a single month of data at a coarse 50-km grid resolution. There is subsequently a clear need to develop tool kits that relinquish this heavy reliance on data sub-sampling or parallelised HPCs for producing scientific data sets. For the latter, this may also begin to bridge the inequality gap between privileged research institutions that have access to such computer resources and those that do not[30].

Over the past decade, the machine learning community has made significant progress in the development of scalable inverse methods[31,32]. These novel methodologies have not yet been widely adopted within the field of Earth observation; however, they are now easily implementable with machine learning libraries, such as GPflow[33] and GPyTorch[34]. Not only do these libraries offer flexibility in terms of constructing GP models, but they crucially provide Graphics Processing Unit (GPU) and batch processing functionalities to speedup linear algebra computations and improve memory handling, respectively. In this study, we present an open-source Python programming library, GPSat, which is built around GPflow, and has been constructed specifically for the purpose of performing efficient spatial (1D or 2D) and spatio-temporal (3D) interpolation of satellite altimetry data. We begin by benchmarking the library against the methodology of G21. For this, we perform joint interpolation of 50-km-gridded CryoSat-2 (CS2), Sentinel-3A (S3A), and Sentinel-3B (S3B) sea ice radar freeboard data, for the 2018/2019 Arctic winter season. We then highlight how GPSat provides considerable speedup relative to G21, and crucially without significant degradation in the derived freeboards. To then demonstrate the scalability of the library, we show examples of interpolating 5-km-gridded radar freeboards, as well as along-track (i.e. footprint-resolution) Sea-Level Anomalies (SLA). It should be noted, however, that while we focus on radar freeboards and SLA here, GPSat is inherently data agnostic and, hence, can be tailored to any field of interest, such as atmospheric weather station data[35].

## Results

### GPSat: local GPs for modelling non-stationary fields

Constructing local models for optimal interpolation is a useful framework with which to both alleviate the computational burden of conventional GP methods, and also flexibly model non-stationary behaviour within climate data sets. Following the approach outlined by G21, for example, if we were to predict the value of radar freeboard at some arbitrary grid point $i$ on day $t$ (illustrated by the white pixel in Fig. 1a), we would first identify all available CS2 and Sentinel-3 (S3) data points that exist within some fixed radius $r$ of location $i$ (white circle in Fig. 1a), and subsequently repeat this selection process for all $\pm\tau$ days surrounding day $t$ (in G21, $r = 300$ km and $\tau = 4$ days). These data points would then form the training set of observations, which are used to both optimise the local GP model, and also make a prediction of radar freeboard at grid point $i$. This process can then be repeated for grid point $i + 1$ (the cyan pixel in Fig. 1a), and for all other sea ice-covered grid points.

Similar to G21, our GPSat library is also based on making predictions from local GP models. A key difference is that in GPSat, we define these local models, hereafter local experts, on a coarse 200-km grid, rather than at every prediction location (white and cyan pixels in Fig. 1b, c). To then accommodate for the distribution of local experts on this coarser grid, we introduce an inference domain, which corresponds to a circular region centred around a given expert location (white and cyan shaded circles in Fig. 1b, c), and in which all predictions are made using this same local expert model. In the case of interpolating gridded radar freeboard, the prediction locations within the inference region correspond to all available grid points for which sea ice concentration is greater than 75%. Meanwhile, in the case of interpolating along-track SLA, the prediction locations are both the sea ice lead and floe locations along the orbital track of interest (black profile in Fig. 1c). Where inference domains from neighbouring expert locations overlap, the predictions can then be combined via a weighted average based on the distance to each expert location. Note that we chose 75% as our sea ice concentration threshold when interpolating gridded radar freeboard, as our training data were only processed at locations with ≥75% sea ice cover[15]. Therefore we do not have observations in the marginal ice zones to validate predictions. While we do not believe that this changes the take-home message of this study, we recognise that appropriate consideration should be made for any operational use case of GPSat, as predictions in these locations will likely carry larger uncertainty due to dynamic ice conditions and uncertainty related to spatial sampling of CS2 and S3 satellites[29].

The G21 methodology can be seen as a limiting case of GPSat in which every prediction location has its own local expert model. In Fig. 1a, we can see how a local expert at every grid point is potentially superfluous given that, in this particular example, 98% of the data points which fall within the cyan training domain are the same as those within the white training domain. We would, therefore, expect to derive very similar GP models for these two locations. Another important distinction to be made is that the G21 approach was written in Python using the NumPy[36] library to perform the linear algebra operations, and SciPy[37] to perform the optimisation procedure. Meanwhile, GPSat is built around GPflow, which itself is based on Tensorflow and has a vast library of modern inference methods for GPs made available to us. Hence, we are able to (1) accelerate linear algebra computations by using GPUs, (2) use automatic differentiation to avoid explicit computations of gradients, and (3) resort to more

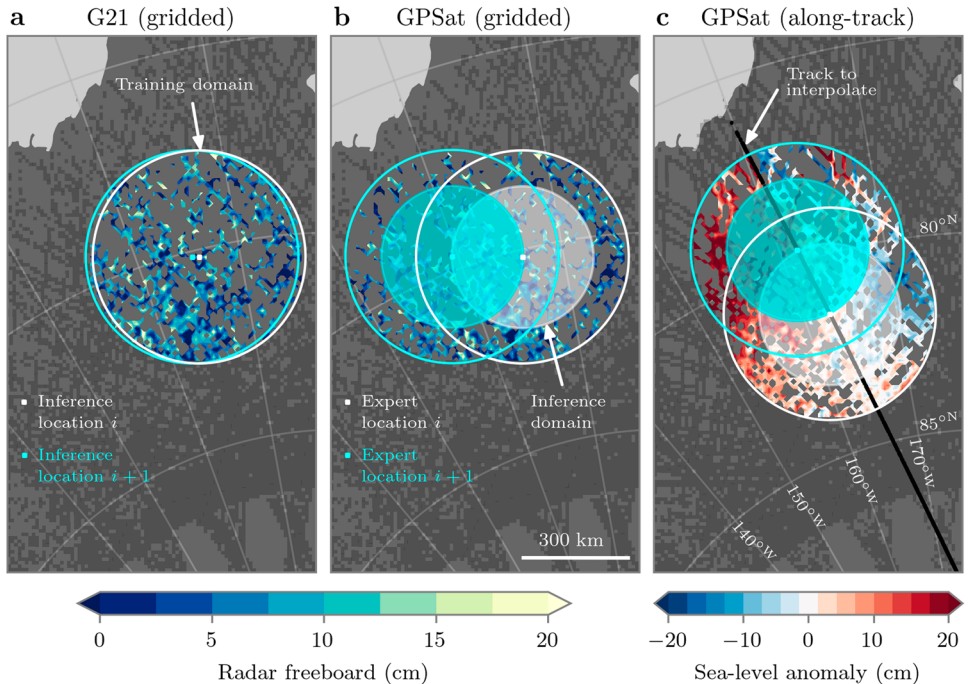

**Fig. 1 | Creating local Gaussian process models for interpolation. a** G21 approach to predicting gridded radar freeboard, where independent models are optimised at every grid point (white and cyan pixels). **b, c** GPSat approach to predicting gridded radar freeboard and along-track sea-level anomalies (SLA), respectively. Local expert models (white and cyan pixels) are optimised using all data within their respective training domains. All prediction locations within the inference domain then share this local expert model. Each figure shows ±4 days of data. Source data for this figure are provided as a Source Data file.

scalable inference techniques than those used in G21, such as inducing point methods[38] (see Supplementary Methods for further details).

## Calibration and runtime performance

In this section, we compare five different approaches for generating daily pan-Arctic fields of 50-km-gridded radar freeboard. The G21 methodology, which we refer to as 'G21N (NumPy CPU)', was validated in their study based on a series of cross-validation tests and comparisons with an independent data set. Therefore we assume that interpolated freeboards from this approach reflect a ground truth. The remaining four methodologies are referred to as: 'G21A (Tensorflow CPU)', 'G21B (Tensorflow GPU)', 'GPSatA (Tensorflow CPU)' and 'GPSatB (Tensorflow GPU)', where the aim is for each of these methods to recover the same freeboards as G21N (or very similar) in a progressively efficient way. In other words, making these comparisons will allow us to quantitatively assess the runtime and prediction differences between three important model configurations: (1) NumPy vs Tensorflow, (2) CPU vs GPU and (3) G21 vs GPSat. Note that all CPU and GPU computations are performed on an EPYC 7H12 64-core AMD processor and an A100-PCIE-40GB NVIDIA graphics card, respectively. Furthermore, we do not use message-passing interface (i.e. parallelised) computing to generate any of the results in this section.

Figure 2 shows an example snapshot of the interpolated freeboards from G21N on January 15 2019, as well as the difference between G21N and each of the configurations described above. Here, we can see that each configuration yields very similar results to G21N, with (rounded) Root Mean Square Differences (RMSD) of 2, 2, 3 and 4 mm, for G21A, G21B, GPSatA and GPSatB, respectively. Some minor freeboard differences may be attributed to differences in numerical optimisation procedures such as gradient computations, and also how linear algebra tricks are handled within both GPflow and G21 (e.g. Cholesky decomposition to compute matrix inverses). Larger freeboard differences generally occur close to the sea ice edge, and also within the Inuit Nunangat region (Canadian archipelago). Notably, these are regions where we typically see large uncertainty in predicted

freeboards (Fig. 2b). In G21, it was suggested that these large uncertainties are likely related to spatial sampling differences of CS2 and S3 satellites (see Supplementary Methods for more information on prediction uncertainty in GP models).

To now check whether the quantitatively similar predictions seen in Fig. 2 are consistent over a longer time period, we compare freeboards for all days between December 1 2018 and April 30 2019. Figure 3 shows the daily RMSD values across this period. Average RMSD values are given as <2 mm for G21A and G21B, and <4 mm for GPSatA and GPSatB. Note that we also show an error for an equivalent GPSatB configuration which uses 600 km separated local experts instead of our standard 200 km implementation. For this test, the average RMSD is equal to 7 mm. From this analysis, we can conclude that the freeboards derived from each method are quantitatively similar to G21A, although we highlight that predictions from GPSat are sensitive to the separation of local experts and so this separation should be tested for each specific use case.

In terms of computational cost, Fig. 4 highlights how the Tensorflow implementation (G21A) scales much better with increasing data size than NumPy (G21N). For the three variations of the G21 methodology presented here, the total compute time to perform pan-Arctic interpolation for all days between December 1 2018 and April 30 2019 amounted to 4539, 586 and 121 h for G21N, G21A and G21B, respectively. We therefore find a 7.7 × speedup moving from NumPy to Tensorflow, and an additional 4.8 × speedup moving from CPU to GPU. Now, given that the G21 methodology is effectively a limiting case of GPSat, the speedup associated with moving to our local expert configuration scales linearly thereafter (e.g. if there are 4000 grid points to predict on a given day, and we divide the Arctic domain into 300 expert locations, then GPSat will be ~4000/300 times faster than G21). For GPSatA and GPSatB, runtimes amounted to 43.5 and 9 h, respectively. This, therefore, corresponds to a total 504 × speedup for 50-km-gridded interpolation between G21N and GPSatB. Additionally, the total runtime for GPSatB with 600-km separated local experts is 2 h, amounting to a 2270 × speedup over G21N. It should be noted,

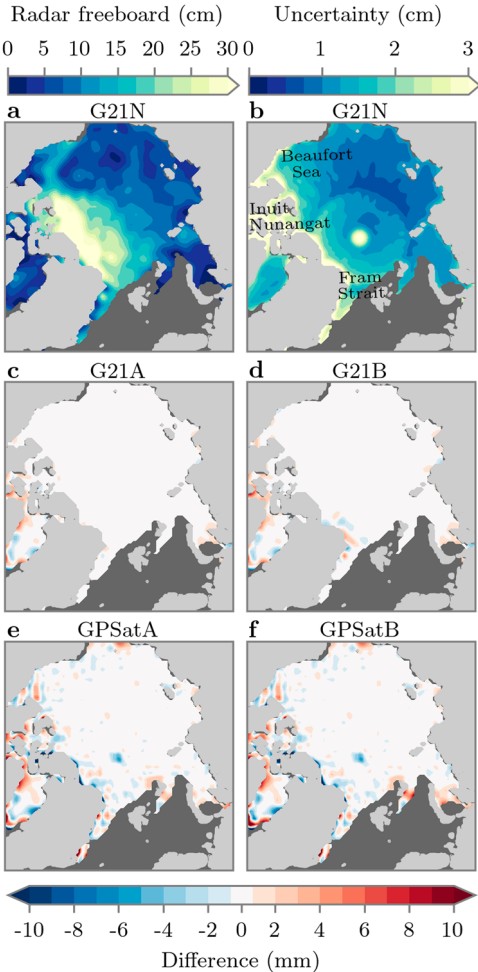

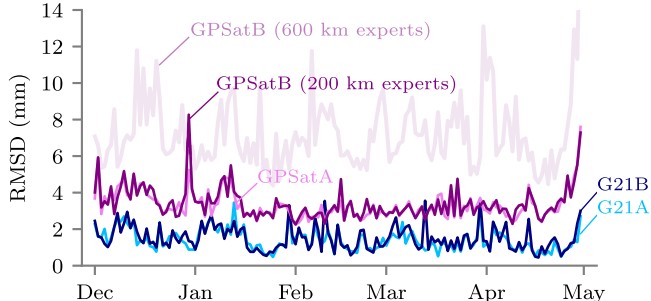

**Fig. 3 | Validation of 50-km-gridded radar freeboard predictions across the 2018/2019 winter season.** Daily root mean square difference (RMSD) of freeboard predictions from G21A (Tensorflow CPU), G21B (Tensorflow GPU), GPSatA (Tensorflow CPU), and GPSatB (Tensorflow GPU); each relative to G21N (NumPy CPU). We also include a GPSatB configuration using 600 km local expert separation. Source data for this figure are provided as a Source Data file.

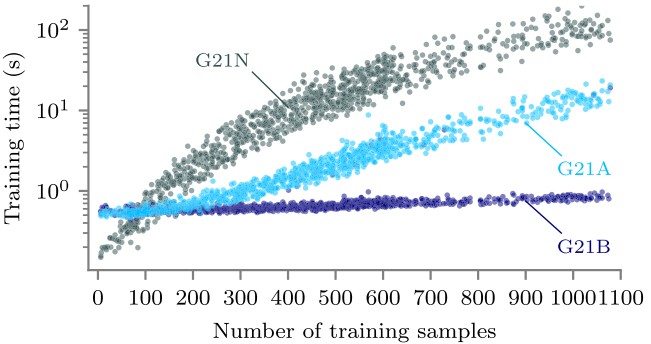

**Fig. 4 | Compute time (logarithmic scale) as a function of training data size, when optimising local Gaussian process models for interpolating 50-km-gridded radar freeboard.** Scatter points correspond to 1000 randomly sampled grid points over the period December 1 2018 to January 31 2019. Shown for each configuration: G21N (NumPy CPU), G21A (Tensorflow CPU) and G21B (Tensorflow GPU). Runtimes of GPSatA and GPSatB scale equivalent to G21A and G21B, respectively. Source data for this figure are provided as a Source Data file.

**Fig. 2 | Validation of 50-km-gridded radar freeboard predictions. a** Interpolated freeboard from G21N (NumPy CPU) on January 15 2019. **b** Uncertainty on interpolated freeboard from G21N on January 15 2019. **c**–**f** Differences in interpolated freeboard, relative to (**a**), for G21A (Tensorflow CPU), G21B (Tensorflow GPU), GPSatA (Tensorflow CPU), and GPSatB (Tensorflow GPU), respectively. Source data for this figure are provided as a Source Data file.

however, that this degree of speedup is likely to improve with increasing grid resolution, given that the scaling with GPU implementation (G21B) is effectively still linear for small enough data sizes. Meanwhile, G21N is already highly non-linear (see Fig. 4). Finally, for data sizes less than ~100 samples, the NumPy CPU implementation is actually faster than both Tensorflow CPU and GPU, which is due to some amount of overhead associated with initialising the GPflow model.

### Scaling potential: 5-km-gridded radar freeboard

The results thus far have been based on optimising local GP models at each grid point or expert location, to learn the spatio-temporal covariance structure of CS2 and S3 radar freeboard observations within a 300-km radius and ±4 days of each location. As discussed in the "Introduction" section, this approach does not generally scale well to large data sets, which, for this particular model configuration, places an upper limit on the achievable grid resolution of ~5 km. At 5 km resolution, the number of training samples at a given expert location can exceed 11,000. Nonetheless, pan-Arctic interpolation of radar freeboard is still achievable in reasonable time with GPSat, at ~32 min per day on a single GPU. As an example, Fig. 5 compares predictions of radar freeboards on December 1 2018, at both 50 and 5 km resolution. We showcase these results here to highlight the potential for GPSat to

provide basin-wide coverage of high-resolution sea ice altimetry data sets, and its potential impacts for climate research. We can see, for example, how the large-scale patterns are qualitatively similar between these two grid resolutions; however, over short spatial scales, the 5 km field naturally exhibits higher variability. In the Fram Strait region east of Greenland, in particular, the 5 km field shows local structure, which is not resolved in the 50 km data. Resolving small-scale variations in ice thickness like this may be crucial for initialising sea ice prediction systems, leading to improved navigability of sea ice-covered regions. When applied to ocean altimetry data, this could also lead to resolved ocean mesoscale or sub-mesoscale eddies. Furthermore, the ability to derive high-resolution uncertainty information on interpolated fields in Fig. 5 is a particularly useful component of GP models. Such information could be utilised in applications ranging from data assimilation[39] to planning the optimal placement of environmental field sensors[40].

To gain a better understanding of the predictive performance of the GPSat model at finer grid resolutions, we now conduct a cross-validation analysis. For each day in December 2018, we use our standard GPSat configuration to interpolate pan-Arctic radar freeboard at 5-km resolution, however we withhold S3A tracks on the prediction day to use for validation. For example, when predicting freeboard on December 15, we train local expert models using all CS2, S3A and S3B data between the 11 and 19, except we withhold all S3A tracks corresponding to the 15. This provides us with a month of data from S3A

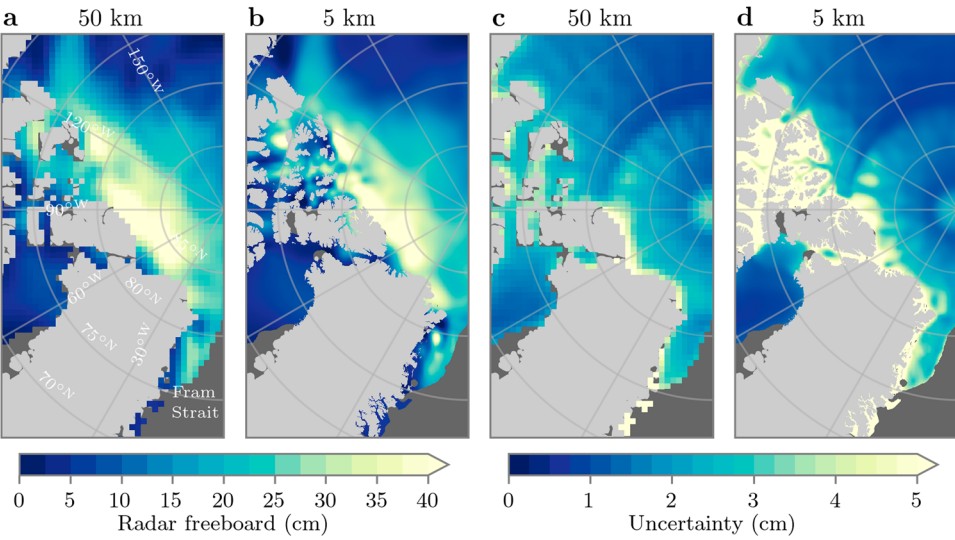

**Fig. 5 | Increasing the spatial resolution of radar freeboard with GPSat. a, b** Radar freeboard predictions at 50 and 5 km, respectively. **c, d** Prediction uncertainty at 50 and 5 km, respectively. Maps correspond to December 1 2018. Source data for this figure are provided as a Source Data file.

with which to validate 5-km GPSat predictions. Figure 6 shows the predictions and associated errors from this analysis. We also include predictions from a linear interpolation approach for comparison, for which we use the same training configuration as GPSat, except we train new linear regression models at each grid point (i.e. we use the G21 configuration rather than our local expert approach). Noticeably the predictions from linear regression are smoother than GPSat, and are unable to resolve small-scale features. Looking again at the Fram Strait region, for example (Fig. 6b vs f), GPSat predicts a localised region of high freeboard, which is not predicted by linear regression. When looking at the spatial error patterns (Fig. 6d vs h), we can see that linear regression shows higher error as a result, which suggests that this is a true feature of the data. Furthermore, the linear approach is not able to fully resolve the tongue of thicker ice in the Beaufort Sea; highlighted by the larger negative spatial errors (Fig. 6c vs g). The inability of linear regression to capture larger positive freeboard values is also emphasised in the 2D histogram in Fig. 6i, where we see maximum predicted freeboards of ~35 cm. Meanwhile GPSat (Fig. 6j) mirrors S3A more closely in this upper range, and is ultimately reflected in the improved RMSD and $R^2$ scores—see also Supplementary Table 1 for a range of RMSD and $R^2$ scores from S3A cross-validation tests which use different training window configurations of GPSat. In this table, we note that a model with a reduced window size of ±200 km/3 days is able to produce the same skill as the ±300 km/4 days configuration shown in Fig. 6, and at over half the runtime cost. In any case, one additional point to note from the histograms in Fig. 6 is that S3A freeboards contain a sizeable number of negative values, which are generally not captured by either linear regression or GPSat. Negative freeboards can occur due to random noise and errors in SLA interpolation[6], and also due to the fact that radar freeboards have not been corrected for waveform propagation delay through snow[41]. Over the December 2018 period we find that the data used to train a given local expert model contain, on average, ten times more positive values than negative, which may suggest why both prediction models generally favour positive values. The fact that the linear regression and GPSat models do not predict negative values (which would be contrary to our physical expectation), provides confidence that the models are not overfitting to the noise in the data.

As a final validation test of 5-km freeboard interpolation, we now compare 3 days of GPSat predictions to three days of gridded airborne data from the NASA Operation IceBridge (OIB) campaign on April 19, 20 and 22 2019. It should be noted that we do not consider OIB

observations in the Fram Strait region south of 81° N, as dynamic ice conditions are likely to result in sampling biases between OIB freeboards and CS2/S3. In Fig. 7, linear regression shows generally good agreement with OIB radar freeboards, with an $R^2$ score of 0.15 (linear correlation of 0.39). Meanwhile, GPSat shows considerable improvements, with an $R^2$ score of 0.43 (linear correlation of 0.66). Similar to the S3A cross-validation analysis, this skill improvement from GPSat appears to be coming, in part, from the fact that GPSat performs better at predicting large positive freeboard values.

## Scaling potential: along-track SLA

Along-track interpolation is a particularly important component of standard sea ice altimetry processing chains. Since the SLA at the position of each floe cannot be measured by the satellite, interpolation offers a means to estimate the SLA at each floe in order to compute sea ice freeboard, and hence thickness. Traditional approaches for interpolating along-track SLA have relied on 1D linear regression[6] or 2D GP regression with heavy data sub-sampling[26]. Arguably, the former case is not sufficient to resolve the complex non-linear behaviour of the ocean surface characteristics, and the latter case can be seen as throwing away valuable training data. Now, given that 5 km is approximately the computational upper limit on the 3D GPSat configuration we use for interpolating gridded radar freeboard (i.e. a ±300 km and 4-day training window), it will not be possible to interpolate along-track SLA with this same approach, where the data resolution is ~300 m. In the "Discussion" section, however, we discuss approximate GP inference methods, which can, in principle, scale to these data sizes, and will be the focus of future GPSat developments. For now, we highlight how 1D and 2D configurations of GPSat can still yield improvements over the standard 1D linear regression approach, and without prohibitive computational expense. In more detail, we make predictions for all CS2 tracks throughout January 2019, where for each individual track, we randomly select 20% of the SLA observations to hold out for validation, and use the remaining 80% for training. In the linear case, we follow the Centre for Polar Observation and Modelling (CPOM) processing sequence for performing 1D along-track SLA interpolation[6]. Specifically, we use all available observations within a 100-km radius of each lead and floe location, in order to predict SLA. For the 1D GPSat implementation, we optimise local expert models at 200 km intervals along each track, using all available CS2 observations (from the same track) that are within a distance of 300 km. For the 2D GPSat implementation, we optimise local expert models at 200 km

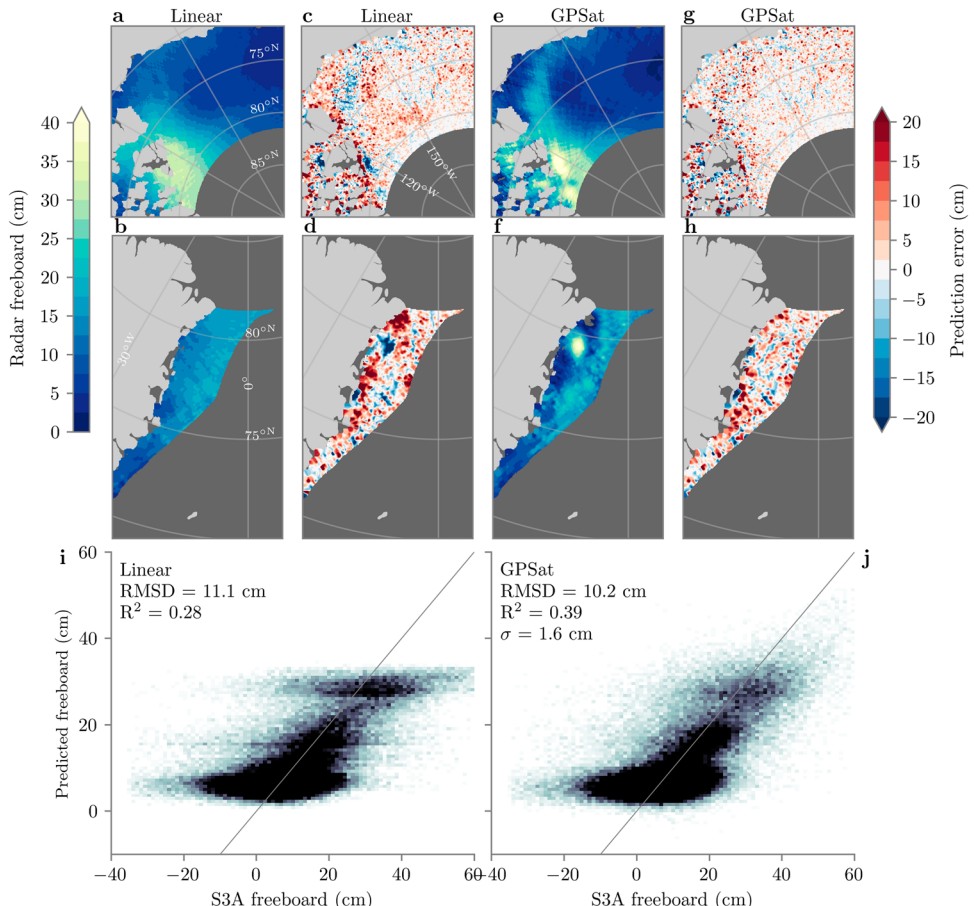

**Fig. 6 | Cross-validation of 5-km-gridded GPSat radar freeboard predictions at Sentinel-3A (S3A) track locations, for December 2018.** **a**, **b** Linear regression predictions in the Beaufort Sea and Fram Strait, respectively. **c**, **d** Linear regression −S3A. **e**, **f** GPSat predictions. **g**, **h** GPSat−S3A. **i**, **j** 2D histograms comparing linear regression and GPSat to held-out freeboards from S3A, for all (pan-Arctic) S3A grid points over December 2018, respectively. The grey line denotes $y = x$, $R^2$ denotes the linear coefficient of determination, and $\sigma$ is the average uncertainty (1 standard deviation) on GPSat predictions. Source data for this figure are provided as a Source Data file.

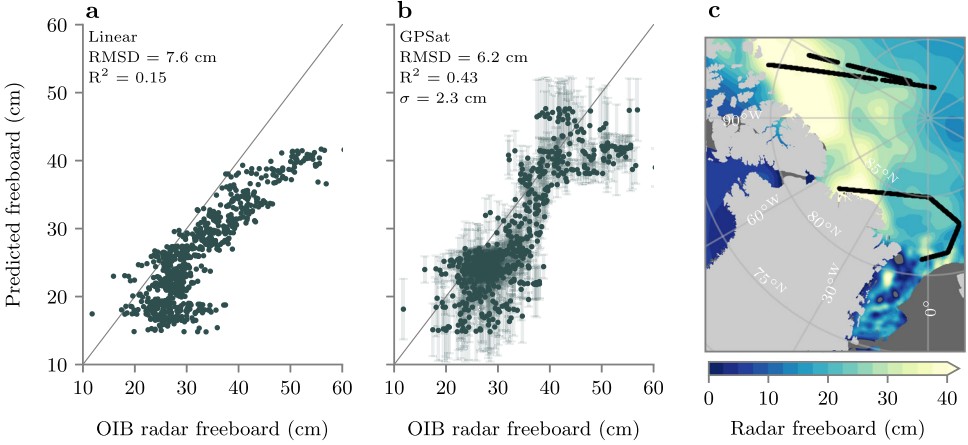

**Fig. 7 | Validation of 5-km-gridded radar freeboard predictions with the 2019 NASA Operation IceBridge (OIB) campaign.** (**a**, **b**) Predictions from linear regression and GPSat, respectively. Error bars in **b** correspond to 1 standard deviation prediction uncertainty, where $\sigma$ is the average standard deviation. The grey line denotes $y = x$ and $R^2$ denotes linear coefficient of determination. **c** The mean interpolated freeboard from GPSat across OIB campaign days, as well as the location of the three OIB transects in black. Source data for this figure are provided as a Source Data file.

intervals along each track, using all available CS2 and S3 observations within a distance of 300-km (recall Fig. 1c). In terms of compute time, linear regression is fast, taking 5 min to interpolate all 1384 tracks across this month-long analysis period. Meanwhile, the GPSat 1D and 2D runtimes amounted to just over 1.5 and 3.5 h on a single GPU, respectively.

In Fig. 8a–c, we can see that both 1D and 2D implementations of GPSat show improvements in predicted SLA relative to linear

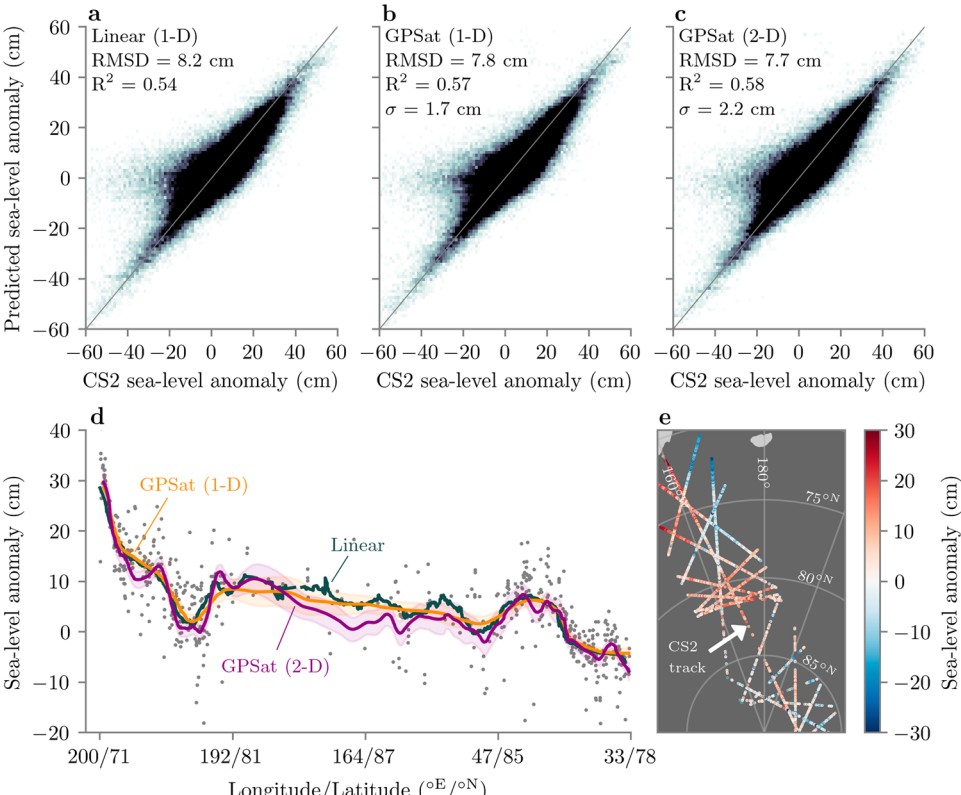

**Fig. 8 | Cross-validation of along-track GPSat sea-level anomaly (SLA) predictions at CryoSat-2 (CS2) track locations, for January 2019. a–c** 2D histograms comparing linear regression, 1D GPSat and 2D GPSat, to held-out SLA observations from CS2, respectively. **d** Example CS2 track on January 15. Scatter points show the held-out SLA samples which were not used in training. Shaded regions reflect 1 standard deviation prediction uncertainty from GPSat. **e** Spatial map showing the location of the CS2 track on January 15, as well as all other CS2 and S3 tracks within a distance of 300-km. Source data for this figure are provided as a Source Data file.

regression, with the 2D implementation showing the highest skill overall. While the improvements in RMSD and $R^2$ metrics appear modest, we emphasise that an RMSD improvement from 8.2 to 7.7 cm is significant, considering the noise on along-track CS2 SLA is estimated to be ~5 cm[14]. It is also worth emphasising here that our GPSat model uses the same GP covariance function as used for radar freeboard interpolation (the Matérn function; see Supplementary Methods for details). Therefore, we do not take into account spatially correlated errors and propagation velocities of SLA; both of which have been shown to improve interpolation estimates[25,26].

In Fig. 8d, we show an example of GPSat and linear regression interpolation for one particular CS2 track. Here we notice that the largest differences between the 1D and 2D GPSat implementations occur north of 81.5°, which is within the S3 polar hole and where CS2 sea ice leads are generally sparse due to the dense ice cover (see Fig. 8e). The 2D implementation therefore utilises information from neighbouring tracks to inform the predictions. Other notable differences in Fig. 8d occur between 200°/71° and 192°/81°, where the 1D GPSat and linear regression approaches are smoother among the cluster of held-out data points. It is not unreasonable, however, to expect complex SLA patterns in this region of the Arctic due to the influence of large-scale ocean circulation patterns such as the Beaufort Gyre[42].

## Discussion

In this study, we introduced GPSat, an open-source Python programming library which is built around the Machine Learning (ML) library, GPflow. We subsequently used observations from the CryoSat-2, Sentinel-3A, and Sentinel-3B radar altimeters, to demonstrate how GPSat enables fast and accurate interpolation of satellite altimetry data. Specifically, we showed that GPSat can be used to generate high spatio-

temporal resolution pan-Arctic gridded radar freeboard data, as well as more accurate sea-level anomaly (SLA) estimates than current operational practices which rely on linear regression. In this section, we discuss potential avenues for improving these results, and the implications of this work in the wider context of climate research.

Throughout this study, we showed examples of using local Gaussian Process (GP) models to interpolate gridded radar freeboard data, as well as along-track (footprint-resolution) SLA. In this configuration, we found that GPSat is over 500 × faster at interpolating 50-km radar freeboard than a previous interpolation scheme developed by ref. 29 (G21). Furthermore, we found that GPSat is able to scale efficiently to a spatial resolution of 5 km. Beyond this, however, runtimes increase considerably and eventually, the number of training data points exceeds the memory limit of a single GPU. Therefore, for along-track SLA interpolation we were required to restrict the size of the training domain for each local GP model to sample data only in 2D space (compared to the gridded freeboard approach which sampled data in 3D, space and time). A solution to this computational problem has already been developed in the literature, and is indeed available within GPflow, and hence GPSat. This is based on using sparse variational GP (SVGP) models, which have been shown to scale to data sizes exceeding $10^6$ samples[38]. In short, SVGP models use all available training data to approximate the data distribution, based on a smaller number of psuedo-inputs known as inducing points (see Supplementary Methods for further details). Arguably, SVGP lends to a more principled solution than simply discarding valuable training data. With the scaling potential of SVGP models, it is conceivable that we could increase the resolution of interpolated gridded radar freeboard to e.g. 1 km or higher. Additionally, we could potentially improve the accuracy of SLA interpolation by also leveraging temporal (3D) information. While we have provisionally tested SVGP on our Arctic sea ice case

here, we have not yet found a configuration which leads to sizeable improvements in predictive skill over GP regression. Further work on this is needed to rigorously evaluate the sensitivity of the SVGP model to the number of inducing points and their initial values. We note here one limitation of the SVGP approach is a general tendency to favour smooth solutions. Therefore, SVGP may result in a loss of precision in areas which are dominated by high-variability signal. In this case, it may be preferable to simply increase the spatial density of local expert models with GPSat, and train using a reduced window size.

A final consideration for increasing the spatial resolution of interpolated fields relates to the intrinsic noise properties of the data. With increasing resolution, both freeboard and SLA data become increasingly noisy. We initially found GPSat prone to over-fitting at 5 km and along-track resolution, which would typically manifest as highly oscillatory predictions over relatively short spatial distances. This over-fitting occurs when local GP models mistake noise for high-variability signal, causing the model to converge on low hyperparameter values of noise variance and correlation lengthscales. While recent work has shown that GP models can be made more robust to noisy data and/or anomalies[43], we opted for a pragmatic solution in this study by applying a lower bound on the noise variance hyperparameter when interpolating fields at finer spatial resolution than 5 km (see Methods). We emphasise that this lower bound should be rigorously tested for each specific use case of GPSat.

More generally, the implications for this work are multi-faceted. In fact, the G21 approach has already been used in recent studies to investigate atmospheric drivers of synoptic-scale radar freeboard variability[23], as well as the potential timing of ice-algal blooms in the Arctic[11]. This highlights the impact which interpolation routines can have on sea ice research. However, due to the computational cost of the G21 approach, these previous studies relied on temporally static (climatology) hyperparameters of local GP models, which ultimately degrades the accuracy of the interpolated freeboard predictions. We propose that GPSat can now bridge this computational gap, and provide accurate high spatio-temporal resolution interpolated fields at a reduced computational cost.

Past studies have also highlighted how uncertainty on along-track SLA is one of the dominant sources of uncertainty in sea ice thickness estimates[6,15]. Therefore, improving SLA interpolation with GPSat has direct consequences for subsequent applications that leverage sea ice thickness observations. For example, the assimilation of CryoSat-2 sea ice thickness data into numerical models has been shown to improve seasonal forecasts of summer Arctic sea ice[39,44]. Improvements in initial conditions coming from more accurate sea ice thickness observations could, therefore, increase seasonal prediction skills across these models. Furthermore, there has been a recent proliferation of studies using ML to derive sub-grid scale climate model parameterisations from data[45–49]. Conventionally, these data come in the form of high-resolution numerical simulations, as observations are considered sparse and/or noisy. We have shown that GPSat can be used to generate de-noised observational data sets above the typical grid resolution of contemporary ocean-sea ice models[50,51]. This could, therefore, provide insights in the field of ML-based model parameterisations, by learning directly from observational data. Finally, high spatio-temporal observations can lead to insights into ocean and sea ice variability. For example, by assessing the impacts of atmospheric forcing on high-resolution sea ice freeboard[23], or ocean circulation patterns[52].

In summary, GPSat represents a step forward for climate data processing routines, one which opens up possibilities for improving understanding and prediction capabilities of key climate processes. Future work will therefore explore the implementation of GPSat over the entire CryoSat-2 record (2010–present), and assess the subsequent downstream impacts on sea ice thickness trends and variability across multiple scales. This will require further sensitivity analysis to determine the generalisation of GPSat to a single satellite altimeter

(Sentinel-3 data are only available after 2018), and additional airborne and ground-based validation tests for the time periods not featured in this present study. Further avenues for computational speedups will also be explored with SVGP models, as well as a parallelised implementation which can leverage multiple GPUs during model training.

## Methods

### Data and processing
**Freeboard.** The sea ice radar freeboard data used in this study are derived from CS2 and S3 Level-0 data, which are processed to Level-1b waveforms using the European Space Agency's Grid Processing on Demand (GPOD) SARvatore service[53]. After processing to Level-1b, radar freeboards are computed according to the Centre for Polar Observation and Modelling (CPOM) sea ice processing sequence[6,15]. The terminology 'radar freeboard' refers explicitly to the waveform echos over sea ice that have not been corrected for the radar propagation delay through the overlying snow layer. As such, these data reflect the height of the radar scattering horizon relative to local sea level, as opposed to the actual height of the sea ice floe, i.e. the 'sea ice freeboard'. We use radar freeboards in this study rather than sea ice freeboards to be consistent with G21. Radar freeboard data are gridded to 50-km and 5-km representations of the Equal-Area Scalable Earth (EASE)[54] grid.

**Sea-level anomaly (SLA).** The SLA data used in this study are also derived from GPOD-processed Level-1b CS2 and S3 waveforms. Following the CPOM processing sequence, SLA corresponds to the instantaneous elevation of the ocean surface (relative to the WGS84 reference ellipsoid) minus the mean sea-surface height, where the mean sea surface was computed as a 2-year climatology of CS2 elevations over the period September 2011 to September 2013.

**Auxiliary data.** In the section "Calibration and runtime performance", we interpolate CS2 and S3 sea ice radar freeboard at all grid point locations that contain sea ice on each day between December 1 2018 and April 30 2019, according to the National Snow and Ice Data Center (NSIDC) NASA Team sea ice concentration data set[55]. These data are provided on a 25 km polar stereographic grid, which we regrid to the 50 km EASE grid using bilinear interpolation. We also regrid these data to the 5 km EASE grid in the section "Scaling potential: 5-km-gridded radar freeboard". We define the presence of sea ice as grid points with a sea ice concentration value ≥75%.

In section "Scaling potential: 5-km-gridded radar freeboard" we also validate interpolation of 5-km-gridded radar freeboard with freeboards derived from the 2019 NASA Operation IceBridge (OIB) airborne campaign[56]. Specifically, we utilise OIB Quick look freeboards from three transects, which were completed on April 19, 20 and 22 2019. OIB measures the elevation of the snow-air interface (snow-freeboard) from an airborne topographic mapper (ATM). Together with a radar-derived snow depth, it is possible to extract the sea ice freeboard along each flight transect. In this study, we compare a theoretical OIB sea ice *radar* freeboard to interpolated CS2 and S3 data from GPSat. Following previous studies, we estimate the OIB radar freeboard by correcting the OIB sea ice freeboard for radar propagation delay through snow[57,58]. To compare OIB freeboards with GPSat, we bin-average the data to the 5-km EASE grid and then apply a five-point smoother to each transect.

### Training local Gaussian process models
This section provides some additional details about the training process for local Gaussian process (GP) models for both G21 and GPSat. It is important to note that in these approaches, when using a 3D training configuration, observations from different days are treated as such, meaning that the GP model derives a local spatio-temporal covariance structure of the observations. The optimisation procedure then

ensures that the hyperparameters that make up the GP model are chosen so that they maximise some objective function, leading to good generalisation well when making predictions at unobserved locations. In both G21 and GPSat, this objective function is the log marginal likelihood[59,60], which is the typical objective for training GPs. When optimising local models for 5-km radar freeboard and along-track SLA interpolation, we set a lower bound of 2.25 cm$^2$ on the likelihood variance hyperparameter. This is to ensure that the models are not over-fitting to the noise in the data (which naturally increases with increasing resolution). In the Supplementary Methods we provide more information on GP theory and how specific methodologies are implemented in GPSat.

## Data availability
The auxiliary sea ice concentration and Operation IceBridge sea ice data sets are both openly available through the National Snow and Ice Data Center (NSIDC) online data portal (https://doi.org/10.5067/MPYG15WAA4WXand 10.5067/GRIXZ91DE0L9, respectively). Cryosat-2 Level-0 data Sentinel-3 Level 1 data were processed to Level-1B (waveforms) using ESA's Grid Processing on Demand (GPOD) service. The GPOD-processed data generated in this study have been deposited in a Zenodo database, version 1.0 [https://zenodo.org/doi/10.5281/zenodo.13218448]. Source data for all figures are also avaiable within the same Zenodo database[61].

## Code availability
The GPSat code is openly available on github https://github.com/CPOMUCL/GPSat. This includes steps of how to configure python environments, as well as examples of how to run the interpolation workflow through Google Colab. A reproducible version of the code can also be found at https://doi.org/10.24433/CO.4875513.v1. Further information can also be found at the GPSat documentation page https://cpomucl.github.io/GPSat/.

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

## Acknowledgements

This research received support through Schmidt Sciences and the Princeton University Library Open Access Fund. C.N. acknowledges support from NERC (#NE/S007229/1) and the UK Met Office (CASE Partnership). M.T. acknowledges support from ESA (#ESA/AO/1-9132/17/NL/MP, #ESA/AO/1-10061/19/I-EF, Clev2er: CRISTAL LEVel-2 procEssor prototype and R&D, SIN'XS: Sea Ice and Iceberg and Sea-ice Thickness Products Inter-comparison Exercise) and NERC (#NE/T000546/1 761 & #NE/X004643/1). S.T. acknowledges support from a Department of Defense Vannevar Bush Faculty Fellowship held by Prof. Andrew Stuart, and by the SciAI Center, funded by the Office of Naval Research (ONR), under Grant Number N00014-23-1-2729.

## Author contributions

W.G., R.M. and S.T. made equal contributions to this study. W.G. beta-tested the GPSat library, and conducted the analysis to generate the figures and write the manuscript. R.M. and S.T. wrote the entirety of the GPSat code and the online documentation. S.T. also wrote the supplementary information for this manuscript. I.R.L. conceptualised the idea for combining CryoSat-2 and Sentinel-3 satellite observations, and was responsible for pre-processing all freeboard and sea-level anomaly data sets. C.N. assisted with beta testing the GPSat library and validation tests. M.P.D. provided intellectual support related to Gaussian process theory and implementation. M.T. conceptualised the idea of applying optimal interpolation to polar altimetry data, assisted with beta testing the GPSat library, as well as providing intellectual support related to polar altimetry. W.G., R.M., S.T., I.R.L., C.N., M.P.D. and M.T. contributed to the development of the text in this article.

## Competing interests

The authors declare no competing interests.
