## [Peer Review File · Nature Communications]

Scalable interpolation of satellite altimetry data with probabilistic machine learningREVIEWER COMMENTS

Reviewer #1 (Remarks to the Author):

This is an excellently written paper presenting a new open-source python library, GPSat, for interpolation of satellite altimetry data using scalable Gaussian Process (GP) techniques. The paper focuses on the interpolation of Altimetry data from the Cryosat-2, Sentinel-3A and Sentinel 3-B missions, to generate daily, Pan-Arctic layers of sea ice thickness (SIT). The use of widely used satellite datasets, as well as the transferability of the GPSat library to other use cases substantially increases the impact of this work. Overall, I believe this paper should be accepted, subject to the completion of additional analysis detailed in the third paragraph of this summary.

The work within this paper builds upon a previous study by the lead author, Gregory et al. (2021), which also used a GP interpolation scheme to generate daily, pan-Arctic, layers of SIT. The key development between the previous paper and this one is the use of scalable GPs which provide impressive computational savings, as evidence within Section 2.2 and Figure 4. The use of scalable GPs also enables the GPSat library to produce SIT layers at 5 km spatial resolution at reasonable timescales compared to the 50 km resolution layers previously produced within Gregory et al. (2021). The agnostic nature of the GPSat library means it can also be used to conduct along-track interpolation of sea-level anomalies- an essential metric to derive prior to calculating ice freeboard.

However, this paper requires more discussion on what these advances in spatial resolution and computational efficiency mean for advancing our knowledge of SIT in the Arctic and the physical processes controlling SIT. It is demonstrated that the GPSat library suffers less from smoothing and can correctly identify areas of higher freeboard compared to traditional linear interpolation models (Figure 6). The 5 km layers also resolve finer-scale features (Figure 5). But there is less emphasis within the paper on what these revised inferred values of SIT tell us about dynamics in the region. For example, do these revised layers tell us that ice is thinner or thicker than previously thought? Are there differences between regions or between areas that are dominated by first verses multi-year ice? Now that GPSat can rapidly produce SIT layers, are there, previously unidentified, annual, seasonal, or spatial trends in SIT? It feels like some of these questions, or analogous questions, should be answerable by these methods and would greatly increase the impact of the study.

There are very few typographical or grammatical errors to correct in the whole paper.

Minor comments:

Introduction: Please add some introductory points on sea ice thickness: Its importance, the reason for using this variable compared with sea ice concentration or extent, what controls thickness and typical ranges etc.

Figure 1: Please provide a scale bar for context. This will also make it easier for the reader when you refer to the circles with different radii.

General figure comment: Please add annotation to one of the figures, or add in an

additional figure, identifying the locations of the place names you mention in the report e.g. lines 211 and 237.

Line 118 – 121: Please provide rationale for only using regions with SIC > 75%. I'm assuming that it is because this ensures ice freeboards are not incorrectly derived in areas of open water, but please confirm. In addition, the use of a relatively high SIC threshold means the majority of the marginal ice zone is removed. Is the performance of the GP tool and other models inferior in these marginal ice zones?

Figure 2b, 5c: Please explain how the GP model handles edge locations where data is not available at very high latitudes. This will help explain the features in these uncertainty layers.

Line 236-237: "The linear approach is not able to fully resolve the 'tongue' of thicker ice in the Beaufort Sea". Linking to the third paragraph in the introduction, it would be great to know whether this is the case on this one date, or whether GPs are in general able to identify sea ice with a larger freeboard in this region, thus indicating that SIT is in general thicker in this region than previously modelled.

Figure 6: It is difficult to discern differences between Figure 6c and 6d. Could additional plots be provided to support the text in Section 2.2.2. This could include zoomed-in insert of the Fram Strait and other regions of interest. Alternatively, an extra plot of absolute error may make the relative magnitudes of error easier to discern.

Line 243 – 246: "Negative freeboards can occur due to random noise and errors in SLA interpolation". Please reflect here or in the discussion on the suitability of these datasets for validation when these uncertainties are present.

Discussion: Please add more information on how these finer resolution, longer time series of SIT could be used in future studies to increase understanding of sea ice conditions and dynamics.

Supplementary information:

Line 23: Change 'related' to 'relate'.

Line 30: "RBF"- Please check as I do not think the Radial Basis Function is explicitly mentioned at an earlier point in the manuscript or supplementary information document. If it isn't, please support the acronym with its full name.

Section 1.2 and Formula 5: I note that you use the same symbol, ℓ , to correspond to the local expert as well as for the length scale. I know ℓ is conventional to use for length scale, but do not know if it is also convention for denoting local experts. If it is, then there is no need to change, otherwise, please consider a different symbol to reduce any confusion (not mandatory).

Reviewer #2 (Remarks to the Author):

NCOMMS-24-19923

Scalable interpolation of satellite altimetry data with probabilistic machine learning

This study proposes a method for scalable interpolation of radar altimetry data using a probabilistic machine learning (ML) method. Specifically, the fusion of CryoSat-2, Sentinel-3A, and Sentinel-3B was achieved by the novel approach GPSat, which is based on the Gaussian Process (GP) algorithm. This study benchmarked Gregory et al. (2021), or G21, and further enhances it with the power of GPflow. The main objective of this study is to simulate the results of G21 using a GPflow-based GPSat, achieving speeds hundreds of times faster by leveraging the modern deep learning library (TensorFlow) and GPU. The overall manuscript is well written with clear motivation, objectives, and contributions. The supporting results also seem necessary and appropriate for the claimed contributions. However, addressing the following issues could further improve this manuscript. My recommendation is a Major revision. Please consider my comments below:

- L37: The full name of TOPEX/Poseidon is missing (Topography Experiment - Positioning, Ocean, Solid Earth, Ice Dynamics, Orbital Navigator).
- L42: Please provide some examples of laser altimeters that use infrared beams. Do all laser altimeters emit infrared beams? What about ICESat-2 with its 532 nm green photon?
- L74-76: Is the result of 36 CPU hours for a single month from G21? Please specify whether this result is from G21 and detail the parallelized HPC (# of CPUs) used in this experiment, if possible. Also, check whether it is 50 km² or a 50x50 km grid.
- L108, 200: In this study, the 3-D window is set to the same parameters as in G21 ($r=300$ km, $t = 4$ days). Is there any option to find the optimal window? It would be beneficial to demonstrate the performance and runtime variance across a diverse range of spatial and temporal scales for interpolation. Relying on a single experimental case may not suffice for a newly suggested algorithm; it is essential to demonstrate the stability of performance relative to window size.
- L334: Are there any factors affecting the cap on spatial resolution other than computational cost? What do the authors expect in terms of uncertainty when the spatial resolution exceeds a 1 km grid? If equipped with larger GPU memory (i.e., modern GPUs with 80GB of memory), is it possible to easily scale up the resolution without significantly losing precision through the sparse GP?
- L335: Please provide some examples of 3-D information for future study.
- L388: The common threshold for binary classifying a sea ice pixel is 15% sea ice concentration. In this study, the authors used 75% as the threshold for sea ice grids. Please provide a brief explanation for this change (same for L118-121).
- As noted at the end of the introduction, GPSat can be extended to other applications. It is recommended to suggest potential example applications.
- Figure 1: Please also indicate what the white and cyan pixels represent in the caption, aside from the manuscript text. What is the black line in the GPSat SLA? Also, add a scale and coordinates to the map (same for other figures with maps).
- Figure 2: I recommend matching the names of G21A-C and GPSatA-B to share the same environment of the framework (NumPy or TensorFlow) and type of processor (CPU or GPU). For example, G21A can be noted as G21N (N for NumPy), G21A & GPSatA share the TensorFlow CPU, and G21B and GPSatB use TensorFlow GPU, respectively. This would help

with better comparison and avoid potential confusion. Please consider this for this figure and the entire manuscript.

- Figure 7: In Fig. 7b, it is hard to recognize error bars for every small point. I recommend adding a mean standard deviation below the R2 metric in this subplot (same for Fig. 8b and 8c).

- Is there any plan for further speedup using parallelization of multiple GPUs? It is expected to facilitate easy scaling with relatively little effort based on modern ML libraries that support multi-GPUs.

Reviewer #2 (Remarks on code availability):

Authors provided a GitHub repository for this study with detailed documentation and examples. Even though I cannot review the details of the code, it appears they have done enough to ensure open access to the code and their experiments.

We would like to thank both reviewers for their excellent feedback, and for their time spent reviewing our work. We feel that their suggested changes have greatly strengthened the manuscript. Please see our responses to each individual comment in orange below. All references to line numbers refer to the tracked-changes manuscript. Please note that we have also made some additional formatting changes to the manuscript (e.g., shortening the abstract) to be in accordance with the journal requirements.

REVIEWER COMMENTS

Reviewer #1 (Remarks to the Author):

This is an excellently written paper presenting a new open-source python library, GPSat, for interpolation of satellite altimetry data using scalable Gaussian Process (GP) techniques. The paper focuses on the interpolation of Altimetry data from the Cryosat-2, Sentinel-3A and Sentinel 3-B missions, to generate daily, Pan-Arctic layers of sea ice thickness (SIT). The use of widely used satellite datasets, as well as the transferability of the GPSat library to other use cases substantially increases the impact of this work. Overall, I believe this paper should be accepted, subject to the completion of additional analysis detailed in the third paragraph of this summary.

The work within this paper builds upon a previous study by the lead author, Gregory et al. (2021), which also used a GP interpolation scheme to generate daily, pan-Arctic, layers of SIT. The key development between the previous paper and this one is the use of scalable GPs which provide impressive computational savings, as evidence within Section 2.2 and Figure 4. The use of scalable GPs also enables the GPSat library to produce SIT layers at 5 km spatial resolution at reasonable timescales compared to the 50 km resolution layers previously produced within Gregory et al. (2021). The agnostic nature of the GPSat library means it can also be used to conduct along-track interpolation of sea-level anomalies- an essential metric to derive prior to calculating ice freeboard.

However, this paper requires more discussion on what these advances in spatial resolution and computational efficiency mean for advancing our knowledge of SIT in the Arctic and the physical processes controlling SIT. It is demonstrated that the GPSat library suffers less from smoothing and can correctly identify areas of higher freeboard compared to traditional linear interpolation models (Figure 6). The 5 km layers also resolve finer-scale features (Figure 5). But there is less emphasis within the paper on what these revised inferred values of SIT tell us about dynamics in the region. For example, do these revised layers tell us that ice is thinner or thicker than previously thought? Are there differences between regions or between areas that are dominated by first verses multi-year ice? Now that GPSat can rapidly produce SIT layers, are there, previously unidentified, annual, seasonal, or spatial trends in SIT? It feels like some of these questions, or analogous questions, should be answerable by these methods and would greatly increase the impact of the study.

There are very few typographical or grammatical errors to correct in the whole paper.

Minor comments:

Thank you for your kind words and constructive feedback! We have responded to each of your comments below, and have updated the manuscript accordingly; providing more discussion on the importance of sea ice thickness and how high spatio-temporal resolution data can lead to new understanding. For the latter, we have made reference to recent studies which adopted the G21 interpolation workflow for impactful sea ice research (see response below). In the revised discussion, we outline a plan for future work which will apply GPSat to the entire CryoSat-2 record (2010-present),

at different spatial resolutions. This future work will require additional sensitivity analysis and validation tests, to assess the generalisation of GPSat to a single satellite altimeter (because Sentinel-3 satellites are only available from 2018-present), and to rigorously evaluate the robustness of derived sea ice thickness trends and variability – we believe this analysis goes somewhat beyond the scope of this present article, which is primarily to document the computational breakthrough achieved with GPSat.

Introduction: Please add some introductory points on sea ice thickness: Its importance, the reason for using this variable compared with sea ice concentration or extent, what controls thickness and typical ranges etc.

Thank you for this suggestion. We have added some further details to the introduction and discussion to provide wider context on the importance of sea ice thickness. Please see L43-47 and L68-74.

Figure 1: Please provide a scale bar for context. This will also make it easier for the reader when you refer to the circles with different radii.

General figure comment: Please add annotation to one of the figures, or add in an additional figure, identifying the locations of the place names you mention in the report e.g. lines 211 and 237.

Thank you for these helpful suggestions. We have revised Figure 1 and subsequent figures to incorporate more location details and explicit labels.

Line 118 – 121: Please provide rationale for only using regions with SIC > 75%. I'm assuming that it is because this ensures ice freeboards are not incorrectly derived in areas of open water, but please confirm. In addition, the use of a relatively high SIC threshold means the majority of the marginal ice zone is removed. Is the performance of the GP tool and other models inferior in these marginal ice zones?

The choice of a 75% threshold was largely related to the fact that the elevation data that we use in this study were only processed above the 75% concentration threshold. Therefore, we do not have data to validate GPSat predictions at locations less than this threshold. We do recognise that predictions in the marginal ice zone will carry larger uncertainty, and have acknowledged this in the revised manuscript (see L151-157 of tracked-changes manuscript).

Figure 2b, 5c: Please explain how the GP model handles edge locations where data is not available at very high latitudes. This will help explain the features in these uncertainty layers.

The GP model is a practical implementation of Bayes' rule. Therefore, we have a **prior** distribution which (in our case) is defined by a zero mean Gaussian with Matérn covariance – see supporting information equation 4. Note that this definition of the prior is independent of any observations. The observations then allow us to construct a **likelihood** distribution, which tells us the probability of having observed these data points, given our GP model. The **posterior** predicted freeboards and uncertainties are then equal to **likelihood x prior**. With more data, the likelihood provides more weight in the prediction, and conversely in the absence of any data, the posterior is equal to the prior. We emphasise that any local expert model which is located within the CS2 polar hole will still have data to form a likelihood distribution, given that the polar hole diameter is approximately 400 km, and our training domain diameter is 600 km. Therefore, the GP predictions here will not exactly equal the prior, but the prior will certainly have more weight than in locations where we have dense observational coverage. We have incorporated this information into the supplementary methods on L31-42.

Line 236-237: "The linear approach is not able to fully resolve the 'tongue' of thicker ice in the Beaufort Sea". Linking to the third paragraph in the introduction, it would be great to know whether this is the

case on this one date, or whether GPs are in general able to identify sea ice with a larger freeboard in this region, thus indicating that SIT is in general thicker in this region than previously modelled.

While the GPSat approach is able to resolve areas of thicker ice, we would be cautious to say that this particular finding is new due to GPSat. For example, if we compute the mean CryoSat-2 freeboard for the month of December (see Figure below (left)), we can see this prominent tongue of thicker ice in the Beaufort Sea. Where we feel that interpolation routines such as GPSat can contribute to new findings is in the analysis of freeboard (and therefore thickness) variability on daily-weekly timescales. For example, G21 showed that having complete daily coverage gives more insights into daily/weekly-scale freeboard variability compared to e.g., a running mean for a given satellite altimeter (we have included a figure from their study below (right), which shows this increased variability). Recent studies have adopted the G21 approach to investigate atmospheric drivers of synoptic-scale freeboard variability (Nab et al., 2023), and also how thickness variability on daily-to-weekly timescales impacts the potential timing of ice-algal blooms (Stroeve et al, 2024). This suggests that these interpolation routines are already impactful for sea ice studies. However, the cost of the G21 approach is prohibitive for high resolution analysis. The studies mentioned above even used a sub-optimal implementation of G21 to derive their analysis (using temporally-static hyperparameters for their local GP models). This ultimately leads to lower accuracy on the derived daily fields. We propose that GPSat can now bridge this computational gap and improve the accuracy of high spatio-temporal resolution freeboard observations, at reduced computational cost. We have expanded on this point in the discussion section (L414-421).

Figure 6: It is difficult to discern differences between Figure 6c and 6d. Could additional plots be provided to support the text in Section 2.2.2. This could include zoomed-in insert of the Fram Strait and other regions of interest. Alternatively, an extra plot of absolute error may make the relative magnitudes of error easier to discern.

Apologies for the lack of clarity in this figure. We have now revised this to show zoomed images of the regions which formulate the majority of the discussion in the main text (Beaufort Sea and Fram Strait).

Line 243 – 246: “Negative freeboards can occur due to random noise and errors in SLA interpolation”. Please reflect here or in the discussion on the suitability of these datasets for validation when these uncertainties are present.

Despite the uncertainties in the observational freeboard data, we argue that these data are still useful for validation. Given that we know negative freeboards are a result of the contribution from various noise factors, we can use this to assess whether a given model is over-fitting. In other words, if a model produces a high R^2 score as a result of predicting negative freeboards, we can conclude that this model is over-fitting as it is not consistent with our physical expectations. We have included a comment on this on L301-304 of the tracked-changes manuscript.

Discussion: Please add more information on how these finer resolution, longer time series of SIT could be used in future studies to increase understanding of sea ice conditions and dynamics.

Following our comment above on the impact of interpolation routines on sea ice analysis, we have expanded the discussion section to make explicit reference to these studies.

Supplementary information:

Line 23: Change 'related' to 'relate'.

Changed.

Line 30: "RBF" - Please check as I do not think the Radial Basis Function is explicitly mentioned at an earlier point in the manuscript or supplementary information document. If it isn't, please support the acronym with its full name.

Thank you, we have revised this to 'squared-exponential' to be in-line with equation 3.

Section 1.2 and Formula 5: I note that you use the same symbol, ℓ , to correspond to the local expert as well as for the length scale. I know ℓ is conventional to use for length scale, but do not know if it is also convention for denoting local experts. If it is, then there is no need to change, otherwise, please consider a different symbol to reduce any confusion (not mandatory).

Thank you for pointing this out. We have replaced all instances of ℓ with κ when referring to the local expert models in the supplementary information.

References

Nab, C., Mallett, R., Gregory, W., Landy, J.C., Lawrence, I.R., Willatt, R., Stroeve, J.C., Tsamados, M. Synoptic variability in satellite altimeter-derived radar freeboard of Arctic sea ice. *Geophysical Research Letters*. 2023.

Stroeve, J.C., Veyssiere, G., Nab, C., Light, B., Perovich, D., Laliberté, J., Campbell, K., Landy, J.C., Mallett, R., Barret, A., Liston, G.E., Haddon, A., Wilkinson, J. Mapping potential timing of ice algal blooms from satellite. *Geophysical Research Letters*. 2024.

Reviewer #2 (Remarks to the Author):

NCOMMS-24-19923

Scalable interpolation of satellite altimetry data with probabilistic machine learning

This study proposes a method for scalable interpolation of radar altimetry data using a probabilistic machine learning (ML) method. Specifically, the fusion of CryoSat-2, Sentinel-3A, and Sentinel-3B was achieved by the novel approach GPSat, which is based on the Gaussian Process (GP) algorithm. This study benchmarked Gregory et al. (2021), or G21, and further enhances it with the power of GPflow. The

main objective of this study is to simulate the results of G21 using a GPflow-based GPSat, achieving speeds hundreds of times faster by leveraging the modern deep learning library (TensorFlow) and GPU. The overall manuscript is well written with clear motivation, objectives, and contributions. The supporting results also seem necessary and appropriate for the claimed contributions. However, addressing the following issues could further improve this manuscript. My recommendation is a Major revision. Please consider my comments below:

Thank you also for your kind words and constructive feedback. We have responded to each of your comments below and have updated the manuscript accordingly. In particular, we have run a suite of sensitivity tests which vary the training domain window size of the local GP models at 5 km resolution. We have also expanded the discussion section to provide more details on the potential challenges associated with high-resolution data and/or sparse GP models.

- L37: The full name of TOPEX/Poseidon is missing (Topography Experiment - Positioning, Ocean, Solid Earth, Ice Dynamics, Orbital Navigator).

Thank you, we have now changed this. Please see L39-40 of tracked-changes manuscript.

- L42: Please provide some examples of laser altimeters that use infrared beams. Do all laser altimeters emit infrared beams? What about ICESat-2 with its 532 nm green photon?

Apologies for this oversight, we have amended this statement in the introduction and included ICESat-2 as an example laser altimeter. Please see L50 and L57-58 of the tracked-changes manuscript.

- L74-76: Is the result of 36 CPU hours for a single month from G21? Please specify whether this result is from G21 and detail the parallelized HPC (# of CPUs) used in this experiment, if possible. Also, check whether it is 50 km² or a 50x50 km grid.

This detail is based on the configuration used in the G21 study, which was 25 parallelised CPU cores (we have included this information in the revised manuscript; L94 tracked-changes). Furthermore, we have changed all references of km² to km. Therefore, when we now refer to a 50 km grid, we mean 50x50 km.

- L108, 200: In this study, the 3-D window is set to the same parameters as in G21 ($r=300$ km, $t = 4$ days). Is there any option to find the optimal window? It would be beneficial to demonstrate the performance and runtime variance across a diverse range of spatial and temporal scales for interpolation. Relying on a single experimental case may not suffice for a newly suggested algorithm; it is essential to demonstrate the stability of performance relative to window size.

Thank you for this suggestion. We have now included a table in the Supplementary Information which repeats the cross-validation analysis from section 2.3 / Fig 6, but for different spatial and temporal window configurations. Note that our upper limit on the window size for 5 km interpolation is 300 km / 4 days. Therefore, our sensitivity tests are for a range of decreasing-sized windows, relative to this upper limit. We do find that we can achieve the same prediction skill to the 300 km / 4 day window with a smaller window of 200 km / 3 days, and subsequently reduce the associated runtime cost. We make a note of this in the tracked-changes manuscript L288-292.

- L334: Are there any factors affecting the cap on spatial resolution other than computational cost? What do the authors expect in terms of uncertainty when the spatial resolution exceeds a 1 km grid? If equipped with larger GPU memory (i.e., modern GPUs with 80GB of memory), is it possible to easily scale up the resolution without significantly losing precision through the sparse GP?

This is certainly an interesting point. In theory, there is no significant overhead when making predictions at higher spatial resolutions (given the same number of data points), since the operation of predicting at

each point in the inference domain is parallelised on the GPU. However, moving to higher spatial resolution does present some other challenges:

- 1) **Precision:** As we mention in the manuscript, the use for sparse GPs at high resolution may be required to alleviate the computational burden associated with having tens of thousands of data points within each training domain. In principle, sparse GPs do not guarantee loss of precision, so long as the underlying ground truth field is smooth, with large lengthscales. When this is not the case it may be prudent to simply increase the density of expert locations, each with a smaller training radius so that one can use a standard GP model to fit the data.
- 2) **Over-fitting:** We do find that as we increase the grid resolution, the freeboard data become increasingly noisy, and that the GP model can often over-fit to this noise (same for SLA). When this over-fitting happens, the model converges to low hyperparameter values of correlation lengthscales and noise variance. Some recent approaches in the literature have attempted to develop GP models which are more robust to noisy data and/or outliers (Altamirano et al., 2023), and we will look to incorporate these developments into GPSat in the future. For this study, we found that placing a lower bound on the noise variance hyperparameter at 5 km grid and along-track resolution was sufficient to prevent over-fitting.

We have incorporated these details into the discussion section (see L397-412 of the tracked-changes manuscript).

- L335: Please provide some examples of 3-D information for future study.

Apologies for the confusion here. We simply mean that we could increase the training window configuration from 2-D to 3-D. In other words, from the implementation shown in section 2.4 where $r=300$ and $t=0$, to one where $t>0$. We have rephrased the start of this paragraph in the discussion section to emphasise what we mean by 2-D and 3-D. Please see L380-383 and L393 tracked changes.

- L388: The common threshold for binary classifying a sea ice pixel is 15% sea ice concentration. In this study, the authors used 75% as the threshold for sea ice grids. Please provide a brief explanation for this change (same for L118-121).

Please see our response to Reviewer 1, who similarly raised this point. Please see also L151-157 of the tracked changes manuscript.

- As noted at the end of the introduction, GPSat can be extended to other applications. It is recommended to suggest potential example applications.

Thank you for this suggestion, we have included an additional example of atmospheric weather station data. Please see L118-119.

- Figure 1: Please also indicate what the white and cyan pixels represent in the caption, aside from the manuscript text. What is the black line in the GPSat SLA? Also, add a scale and coordinates to the map (same for other figures with maps).

We have revised Figure 1 and explicitly labeled each of the important features.

- Figure 2: I recommend matching the names of G21A-C and GPSatA-B to share the same environment of the framework (NumPy or TensorFlow) and type of processor (CPU or GPU). For example, G21A can be noted as G21N (N for NumPy), G21A & GPSatA share the TensorFlow CPU, and G21B and GPSatB use TensorFlow GPU, respectively. This would help with better comparison and avoid potential confusion. Please consider this for this figure and the entire manuscript.

Thank you for this suggestion. We have replaced all instances of G21A with G21N, and similarly changed G21B and G21C to G21A and G21B, respectively.

- Figure 7: In Fig. 7b, it is hard to recognize error bars for every small point. I recommend adding a mean standard deviation below the R2 metric in this subplot (same for Fig. 8b and 8c).

Apologies, upon review we realised a mistake in the display of the error bars in Fig 7. We have revised this and have also included the mean standard deviation statistic for reference (same for Figs 6 and 8).

- Is there any plan for further speedup using parallelization of multiple GPUs? It is expected to facilitate easy scaling with relatively little effort based on modern ML libraries that support multi-GPUs.

This is indeed a plan for future development and should be relatively straightforward as you say. We have provided a brief statement regarding this in the discussion section – L446-448.

Reviewer #2 (Remarks on code availability):

Authors provided a GitHub repository for this study with detailed documentation and examples. Even though I cannot review the details of the code, it appears they have done enough to ensure open access to the code and their experiments.

Thank you for checking this.

References

Altamirano, M., Briol, FX. & Knoblauch, J. Robust and conjugate Gaussian process regression. Preprint at <https://arxiv.org/abs/2311.00463> (2023).

REVIEWERS' COMMENTS

Reviewer #1 (Remarks to the Author):

Thank you for responding to all my comments. I believe the rebuttals are sufficient, and look forward to the subsequent paper applying this tool at large scale as I believe the results may be very informative.

I did not identify any typographic errors in the revisions, and subsequently suggest this paper is accepted for publication.

Reviewer #2 (Remarks to the Author):

NCOMMS-24-19923A

Scalable interpolation of satellite altimetry data with probabilistic machine learning

I appreciate the time and effort the authors have put into the successful revision. I have read the comments and corresponding updates on the manuscript, which is enough for me to agree with. I believe this paper is now ready to move forward for publication.

Reviewer #2 (Remarks on code availability):

They provided their full code on the interactive platform with a well-organized structure. I think this is perfect.